# Physical Activity and Frailty Are Impaired in Older Adults with Benign Paroxysmal Positional Vertigo

**DOI:** 10.3390/jcm13247542

**Published:** 2024-12-11

**Authors:** Sara Pauwels, Nele Lemkens, Winde Lemmens, Kenneth Meijer, Wouter Bijnens, Pieter Meyns, Raymond van de Berg, Joke Spildooren

**Affiliations:** 1Faculty of Rehabilitation Sciences, REVAL-Rehabilitation Research Centre, Hasselt University, 3590 Diepenbeek, Belgium; pieter.meyns@uhasselt.be (P.M.); joke.spildooren@uhasselt.be (J.S.); 2Department of Otorhinolaryngology and Head and Neck Surgery, School for Mental Health and Neuroscience, Faculty of Health Medicine and Life Sciences, Maastricht University Medical Centre, 6229 Maastricht, The Netherlands; raymond.vande.berg@mumc.nl; 3Department of Otorhinolaryngology, Head and Neck Surgery ZOL Hospital, 3600 Genk, Belgium; nele.lemkens@kno.be (N.L.); winde.lemmens@kno.be (W.L.); 4Department of Nutrition and Movement Sciences, NUTRIM Institute of Nutrition and Translational Research in Metabolism, Maastricht University, 6229 Maastricht, The Netherlands; kenneth.meijer@maastrichtuniversity.nl; 5Research Engineering (IDEE), Maastricht University, 6229 Maastricht, The Netherlands; wouter.bijnens@maastrichtuniversity.nl

**Keywords:** older adults, BPPV, physical activity, frailty, dizziness, well-being

## Abstract

**Background/Objectives**: Benign Paroxysmal Positioning Vertigo (BPPV), diagnosed in 46% of older adults with complaints of dizziness, causes movement-related vertigo. This case-control study compared physical activity, frailty and subjective well-being between older adults with BPPV (oaBPPV) and controls. **Methods**: Thirty-seven oaBPPV (mean age 73.13 (4.8)) were compared to 22 matched controls (mean age 73.5 (4.5)). Physical activity was measured using the MOX accelerometer and the International Physical Activity Questionnaire. Modified Fried criteria assessed frailty. The Dizziness Handicap Inventory, Falls Efficacy Scale, and 15-item Geriatric Depression Scale assessed subjective well-being. A post-hoc sub-analysis compared all variables compared between frail oaBPPV, robust oaBPPV and robust controls. Significance level was set at α = 0.05. **Results**: oaBPPV were significantly less physically active and were more (pre-)frail (*p* < 0.001) compared to controls. They performed significantly less active bouts (*p* = 0.002) and more sedentary bouts (*p* = 0.002), and a significantly different pattern of physical activity during the day. OaBPPV reported significantly less time in transportation activities (*p* = 0.003), leisure (*p* < 0.001), walking (*p* < 0.001) and moderate-intensity activities (*p* = 0.004) compared to controls. Frail oaBPPV were even less active (*p* = 0.01) and experienced more fear of falling (*p* < 0.001) and feelings of depression (*p* < 0.001) than robust oaBPPV and controls. **Conclusions**: BPPV can induce a vicious cycle of fear of provoking symptoms, decreased physical activity, well-being and consequently frailty. It is also possible that frail and less physically active older adults have an increased prevalence of BPPV. Post-treatment follow-up should assess BPPV, frailty and physical activity to determine if further rehabilitation is needed.

## 1. Introduction

Dizziness and unsteadiness are a common problem among older adults, limiting their daily activities. Benign Paroxysmal Positioning Vertigo (BPPV) is diagnosed in 46% of older adults with complaints of dizziness [1]. BPPV is a vestibular disorder caused by dislodged otoconia that migrate into the semicircular canals. Typical symptoms of attacks of vertigo, nystagmus, nausea and imbalance when the head is moved in the plane of the affected canal (e.g., when lying down or rolling in bed, tilting the head when looking upwards) [2]. However, older adults often present atypical symptoms [3], such as lightheadedness and imbalance between attacks, or even no dizziness at all [4], leading to a delayed diagnosis [5].

It is known that BPPV can negatively affect quality of life [6], postural control [7,8], and increases the odds and fear of falling in older adults [9]. This may lead to a vicious cycle of more fear of falling, limiting their physical activities.

The benefits of regular physical activity in older adults are well established. It is essential for psychological health and the prevention of chronic diseases such as heart disease, type 2 diabetes and frailty [10]. Frailty is defined by Fried et al. a clinical syndrome, diagnosed if at least two of the following criteria are present: (I) unintentional weight loss, (II) exhaustion, (III) weakness, (IV) slowness and/or (V) low physical activity [11]. Fried’s criteria are mostly focused on physical frailty. Frailty is as a high-risk state predictive for adverse health outcomes, leading to a six-fold increased mortality-rate over three years [11].

Previous studies found that adults with BPPV reported significantly less physical activity than controls [12], and that women who were not regularly physically active were 2.62 times more likely to have BPPV [13]. According to Martelucci et al. the lack of resuming daily physical activities after the gold standard treatment with repositioning maneuvers is associated with more residual dizziness after treatment [14]. However, none of these studies were focused on older adults, where a lack of physical activity can interact with frailty, psychological aspects, and consequently affect healthy aging. Also, physical activity was assessed via a questionnaire, which is prone to recall bias and social desirable answers [15]. Previous research has already demonstrated this recall bias in patients with vestibular disorders [16].

Therefore, this study aimed to compare both objective and subjective measures of physical activity, frailty and subjective well-being between older adults with BPPV (oaBPPV) and without BPPV (controls). It was expected that oaBPPV would be less physically active and would have a decreased subjective well-being compared to controls. These findings could be more pronounced in frail older adults with BPPV (oaBPPV_frail_).

## 2. Materials and Methods

This study is approved by the ethical committees of Hospital Oost-Limburg, Genk (ZOL Genk) and Hasselt University (B3712021000013) and complied with the declaration of Helsinki. It is registered in ClinicalTrials.gov (NCT03526653).

### 2.1. Participants

As part of a larger prospective study, community-dwelling older adults diagnosed with BPPV were recruited at the department of Otorhinolaryngology of Hospital Oost-Limburg, Genk (ZOL Genk), between September 2021 and July 2023. When diagnosed with BPPV, they were screened for eligibility criteria. Inclusion criteria for the larger study were: (1) persons ≥65 years old, (2) able to stand independently for at least 30 s, (3) able to walk (with or without) walking aid for at least 10 m, (4) diagnosed with posterior semicircular canal BPPV or lateral semicircular canal BPPV (geotropic or apogeotropic variant) and not yet treated for the current episode of BPPV. Exclusion criteria were: (1) being unable to understand and follow simple instructions (e.g., due to severe dementia), (2) having contra-indications for the diagnostic maneuver or caloric irrigation test (e.g., perforation of the tympanic membrane), (3) having an evolutionary disorder of the central nervous system (e.g., Parkinson’s disease), (4) being in rehabilitation for an orthopedic or cardiovascular incident, (5) having a resolution of BPPV before data collection was completed.

An age-, weight- and height-matched control group of older adults (≥65 years) without BPPV was also recruited via organizations for seniors and the network of the researchers. The partners of the participating patients were also invited to participate in the control group. With exception for the presence of BPPV, the same eligibility criteria applied for the control group.

### 2.2. Study Design 

Informed consent was obtained from all individual participants included in the study. After given informed consent and a confirmed diagnosis of BPPV, assessed with video frenzel goggles (VisualEyes™ 505 Video Frenzel system Interacoustics, Middelfart, Denmark) by a trained audiologist or SP, demographic data were collected (Figure 1). The video frenzel goggle prevents fixation, which can influence characteristics of nystagmus, while providing the examiner a magnified view of the eyes and the ability to review the nystagmus without needing to repeat the test [17]. Demographic data included age, weight, height, use of walking aid, sleeping pattern, comorbidities and number of medications. The duration of their complaints of BPPV was questioned and classified into “some days”, “several weeks” or “several months”.

The Dizziness Handicap Inventory (DHI) [18], Falls Efficacy Scale (FES-I) [19], and 15-item Geriatric Depression Scale (GDS-15) [20] were filled out at home to assess the impact of dizziness on daily activities, fear of falling and feelings of depression, respectively. The DHI is a 25-item questionnaire that quantifies the impact of dizziness on daily activities with a physical (7 items), emotional (9 items) and functional (9 items) subscale and a minimum (i.e; best) score of 0 and maximum (i.e., worst) score of 100 [18]. The DHI has an excellent test-rest reliability and significant content validity [21]. The FES-I is a 16-item questionnaire that quantifies an individuals’ concern of falling on a 4 point Likert scale, with 1 as not concerned and 4 as very concerned [19]. The FES-I has excellent test-retest reliability and validity in vestibular disorders [22]. The GDS-15 is a “yes” or “no” questionnaire screening for depression in older adults with a score from 0 (no depression) to 15 (severe depression) [20]. The GDS-15 as a 86% sensitivity and 79% specificity to detect depression in older adults [23].

To assess frailty, Fried criteria [11], adjusted as proposed by Avila-Funes et al. [24], were used due to the feasibility within the larger protocol. Participants with three or more frailty components were considered “frail”, those with one or two criteria were “prefrail”, and those with none were considered “robust”.

Physical activity was objectively assessed with the MOX accelerometer and MOXS1WO software version 1.1.0 (MOX; Maastricht Instruments BV, Maastricht, NL) [25]. The device was placed at the right upper leg of the participants with a specifically designed plaster. Due to the body placement and waterproof design, it was unnecessary to remove the accelerometer during showering or sleep. Participants were instructed to wear the MOX at least four consecutive days including at least one weekend day. An algorithm analyzes raw acceleration data from the MOX to classify physical activity into five categories: sedentary (activities spend in a reclined/ sitting or lying position) [26] behavior, standing, light activity (LPA), moderate activity (MPA), and vigorous activity (VPA). It uses a decision tree to separate the data into sedentary, standing, or dynamic behaviors. The data is split into one-second segments, and activity levels are measured as counts per second. Based on these counts, segments are classified as sedentary or dynamic. Dynamic segments with up to 8 counts per second are labeled LPA, 8–16 counts as MPA, and more than 16 counts as VPA [25].

The following parameters were derived:Mean sedentary, standing and dynamic minutes/day. Dynamic minutes/day was also subdivided into mean minutes LPA/day, mean minutes MPA/day and mean minutes VPA/day. Percentages of these three classifications over dynamic time were also calculated.Mean number of postural transitions (from sedentary to upright)/day.Mean number of dynamic bouts of (I) ≥5–<10 min/ day and (II) ≥10 min/day. The number of participants with dynamic bouts ≥10 min and mean duration (minutes) of their bouts ≥10 min were calculated. Although physical activity of any bout duration is associated with improved health outcomes [27,28], this cut-off was set to be in agreement with the ‘International Physical Activity Questionnaire’ [29], a measurement of subjective physical activity.For sedentary behavior, the mean number of bouts/day and mean bout duration (minutes) of bouts ≥30 min was calculated. These variables provide more insight in the alternation of physical activity and sedentary behavior during the day (e.g., participants may have the same number of dynamic and sedentary minutes/ day, but engage in multiple short bouts/day or fewer long bouts/day).The intensity of physical activity was summed up per hour for 24 h, and the mean for each hour over four days was calculated to compare the mean distribution of physical activity intensity during the day between both groups.

Maximum 7 days after recruitment (T0) oaBPPV completed the assessments (T1). The questionnaires for subjective well-being were checked for completeness. The Montreal Cognitive Assessment (MOCA) [30] was used to screen for cognitive impairment. The MOCA is a screening tool for mild cognitive impairment, evaluating visuospatial skills, attention, language, abstract reasoning, delayed recall, executive function, and orientation. The MOCA has a high content validity and sensitivity for mild cognitive impairment [30].

With the long-form International Physical Activity Questionnaire (IPAQ) [31]. The IPAQ is a 27-item self-reported measure of duration and frequency of physical activity (work-related, transportation, household/gardening and leisure-time activities) and sedentary behavior (time spent sitting) of the past seven days to assess subjective physical activity. The IPAQ has been proven reliable to inquire physical activity among older adults [32]. The following outcome parameters were derived:The classification level of physical activity (low, moderate or high) based on the total volume and the number of days of physical activity.Each type of activity was weighted by its energy requirements and defined in multiples of the metabolic resting rate (METs) to calculate a score in MET-minutes [31] MET-minutes was computed by multiplying the MET-score of an activity by the minutes performed. A total MET-minutes/week was calculated for each of the four domains, walking, moderate-intensity activities and vigorous-intensity activities.Mean sitting minutes/day were calculated.

### 2.3. Statistics

Data analysis and graph creation were done using the IBM SPSS statistics software (v25.0 for Windows, SPSS Inc., New York, NY, USA) and GraphPad Prism 10 (GraphPad Software, San Diego, CA, USA).

As data on physical activity in oaBPPV lack in the literature, no a priori sample size calculation was performed for the current analysis. Therefore, a sensitivity power analysis for the current sample sizes at 80% power and α = 0.05 was performed, using G*Power (Version 3.1.9.6). The required Cohen’s d are presented in Appendix A.

Data was checked for normality with Shapiro-Wilk tests. Significant outliers were identified with Tukey’s method and excluded if necessary based on consensus. Continuous data was analyzed with an unpaired t-test and Mann-Whitney U test for normal and non-normal distributed data, respectively. Categorial data were analyzed with the Pearson Chi-square test. Effect sizes for non-parametric tests were calculated as Cohen’s d according to Fritz et al. [33]. To analyze differences in mean distribution of physical activity during the day, random-intercept linear mixed models were used with time added as a quadratic variable (time^2^). To get more insight in differences in intensities at each hour, Mann-Whitney U test with Bonferroni correction was applied. Normally distributed data are expressed as mean (SD), non-normally distributed data as median (minimum-maximum).

To correct for multiple comparisons, the Holm-Bonferroni correction [34] was applied within following groups: subjective well-being (DHI and subscales, FES-I and GDS-15), frailty (total score and subscores), MOX (average minutes/day, number of bouts and bout duration) and IPAQ (categorical and continuous score).

As oaBPPV were significantly more (pre-)frail compared to controls, post-hoc sub-analysis was conducted that compared all variables between frail oaBPPV (oaBPPV_frail_), robust oaBPPV (oaBPPV_robust_) and robust controls (controls_robust_) to assess the importance of frailty. Continuous data was analyzed with a one-way ANOVA and Kruskall Wallis test for normal and non-normal distributed data, respectively. A correction for multiple post-hoc comparisons were conducted with Tukey after one-way ANOVA and Bonferroni after Kruskall Wallis. All other statistics were similar as described above. Results and *p*-values are presented in the Appendix A.

## 3. Results

Thirty-seven oaBPPV (23 females, mean age 73.1 (4.8)) were compared to 22 controls (12 females, mean age 73.5 (4.5)) (for the selection process of the participants, see Figure 1). Results on characteristics are presented in Table 1. Groups were matched for gender (*p* = 0.77), age (*p* = 0.77), weight (*p* = 0.25) and height (*p* = 0.16), but significantly different in number of medications (*p* = 0.007) and cognition according to MOCA (*p* < 0.001). All walking aids were already in use prior to the presence of BPPV.

Post-hoc sub-analyses revealed that oaBPPV_frail_, oaBPPV_robust_ and controls_robust_ were equally matched, but oaBPPV_frail_ and oaBPPV_robust_ performed significantly worse on the MOCA (*p* = 0.008) then controls_robust_. OaBPPV_frail_ had a significantly higher number of medications (*p* = 0.01) compared to oaBPPV_robust_ and controls_robust_ (Appendix A).

### 3.1. Frailty 

Significantly more oaBPPV were frail or pre-fail compared to controls (*p* < 0.001) (Table 2). They experienced more self-reported exhaustion (*p* < 0.001) (“How often did you feel that everything you did required effort over the past week?” and “How often did you feel unable to get going over the past week?”), slowness (*p* = 0.001) (gait speed on 10 m walk test) and weakness (*p* = 0.005) (“Do you experience difficulties rising from a chair?”). No differences were found in physical inactivity (*p* = 0.14) (“Do you regularly engage in physical activities such as walking, gardening, or sports?”) and unintentional weight loss (*p* = 0.19) (“Have you unintentionally lost > three kilograms last year?” or body mass index ≤ 21 kg/m^2^).

### 3.2. Subjective Well-Being

The DHI (total score and physical, emotional and functional subscale) was significantly higher in oaBPPV compared to controls (32 (8–74) vs. 0 (0–10)). OaBPPV also experienced significantly more fear of falling and feelings of depression according to the FES-I (25 (2–53) vs. 17.5 (8–31)) and GDS-15 (2 (0–13) vs. 1 (0–4)) (Figure 2). Cohen’s d of GDS-15 was too small (<0.69) according to sensitivity analyses, indicating that power is less than 80%.

Post-hoc sub-analyses revealed that total score (*p* < 0.001) and the functional subscale (*p* < 0.001) of the DHI were significantly different between the three groups (oaBPPV_frail_ > oaBPPV_robust_ > controls_robust_). The emotional (*p* < 0.001) and physical subscales (*p* < 0.001) were significantly higher in oaBPPV_frail_ and oaBPPV_robust_ compared to controls_robust_. The FES-I (*p* < 0.001) also significantly differed between the three groups (oaBPPV_frail_ > oaBPPV_robust_ > controls_robust_). OaBPPV_frail_ experienced significantly more feelings of depression (*p* < 0.001) compared to oaBPPV_robust_ and controls_robust_ (Appendix A).

### 3.3. Objective Physical Activity

OaBPPV had significantly fewer dynamic minutes/day compared to controls. They performed fewer low-, and moderate-intensity activities, while vigorous intensity did not significantly differ (Figure 3). Within their dynamic time, oaBPPV spend 69.5% of their time in low-intensity activities, while 29.9% and 0.5% was spent in moderate and vigorous activities, respectively. Controls spend 52.8% of the dynamic time engaged in low-intensity activities, while 67.3% and 17.5% was spend in moderate and vigorous activities, respectively. There was a trend towards more sedentary minutes/day in oaBPPV, standing minutes/day did not significantly differ. There was a trend (*p* = 0.04, d = 0.49) towards fewer postural transitions in oaBPPV (185.7 (89.8–519) vs. 249.6 (104.3–419)).

Post-hoc sub-analyses revealed that oaBPPV_frail_ had significantly fewer dynamic minutes/day (*p* = 0.01) compared to oaBPPV_robust_ and control_robust_ and engaged significantly less minutes/day in moderate (*p* = 0.003) and vigorous (*p* = 0.007) physical activities. There was a trend (*p* = 0.02) towards more sedentary minutes/day in oaBPPV_frail_ compared to oaBPPV_robust_ and control_robust._ The number of postural transitions (*p* = 0.34) did not significantly differ between groups (Appendix A).

OaBPPV performed significantly fewer dynamic bouts ≥10 min. They also performed significantly less bouts of 5–10 min, but Cohen’s d was too small (<0.71), indicating that power is less than 80%. Only 28% of oaBPPV performed dynamic bouts of ≥10 min compared to 68.2% of the controls (*p* = 0.002). The mean duration of those ≥10 min-bouts did not significantly differ.

OaBPPV performed significantly more sedentary bouts of ≥30 min. The mean duration of those ≥30 min-bouts did not significantly differ (Table 3).

Post-hoc sub-analyses revealed that oaBPPV_frail_ and oaBPPV_robust_ had significantly more sedentary bouts (*p* = 0.01) compared to control_robust_. There was a trend (*p* = 0.04) towards less dynamic bouts of 5–10 min in oaBPPV_frail_ compared to control_robust_ (Appendix A). The number of dynamic bouts ≥10 min (*p* = 0.08), bout duration of dynamic bouts ≥10 min (*p* = 0.16) and bout duration of sedentary bouts ≥30 min (*p* = 0.4) did not significantly differ.

A significant difference in mean distribution of physical activity was found between oaBPPV and controls (F_1,52_ = 5.98; *p* = 0.02), during the day (F_1,1240_ = 1053.5; *p* < 0.001) and for the quadratic variable time^2^ (F_1,1240_ = 941; *p* < 0.001). Post-hoc comparison of mean intensity at each hour revealed a trend towards an increased intensity at 12 a.m. (*p* = 0.01) and a significantly increased intensity at 1 a.m. (*p* = 0.001) in oaBPPV. In the morning, their intensity was significantly decreased at 7 a.m. (*p* = 0.002), 8 a.m. (*p* < 0.001) and 9 a.m. (*p* < 0.001), and they had a trend towards a decreased intensity at 10 a.m. (*p* = 0.004). In the afternoon, oaBPPV had a significantly decreased intensity at 2 p.m. (*p* < 0.001) (Figure 4).

Based on visual inspection, there appeared to be a 1 h time-shift between oaBPPV and controls (Figure 4A), causing the significant differences in intensity. It appeared that oaBPPV started their day at 7 a.m., took a break between 12 a.m. and 2 p.m. and ended their day at 1 a.m.. Controls started their day at 6 a.m., took a break between 11 a.m. and 1 p.m. and ended their day at 12 a.m.. Therefore, analyses was re-done without the first hour of oaBPPV and last hour of controls (Figure 4B). A significant difference in mean distribution of physical activity was again found between oaBPPV and controls (F_1,52_ = 6.12; *p* = 0.02), during the day (F_1,1186_ = 1073.94; *p* < 0.001) and for the quadratic variable time^2^ (F_1,1186_ = 950.4; *p* < 0.001) Post-hoc comparison of mean intensity at each hour revealed a trend toward a decreased intensity at 9 a.m. (*p* = 0.04), 10 a.m. (*p* = 0.04), 12 p.m. (*p* = 0.05), 1 p.m. (*p* = 0.02), 2 p.m. (*p* = 0.01), 3 p.m. (*p* = 0.02), 7 p.m. (*p* = 0.02) and 11 p.m. (*p* = 0.02) in oaBPPV.

In the post-hoc sub-analyses, there was a significant difference in mean distribution of physical activity between oaBPPV_frail_, oaBPPV_robust_ and controls_robust_(F_1,38_ = 4.47; *p* = 0.02) during the day (F_1,941_ = 856.07; *p* < 0.001) and quadratic variable time^2^ (F_1,941_ = 941; *p* < 0.001) Post-hoc comparison of mean intensity at each hour revealed a significant decreased intensity in oaBPPV_frail_ at 8 a.m. compared to oaBPPV_robust_ and control_robust_, and a significantly decreased intensity in oaBPPV_frail_ and oaBPPV_robust_ at 9 a.m. compared to control_robust_ (Appendix A).

### 3.4. Subjective Physical Activity 

OaBPPV reported significantly less MET-minutes/week for transport, leisure activities, walking and moderate physical activity compared to controls. There was a trend towards decreased MET-minutes/week for work in oaBPPV. The MET-minutes for household, sitting and vigorous physical activities did not significantly differ (Table 3).

There was a trend towards a different physical activity-classification between oaBPPV and controls. Twelve oaBPPV were categorized as low physically activity, one as moderate and twenty-four as vigorous physically active. Two controls were classified as low physically active, all others were considered to be vigorous physically active.

The post-hoc sub-analyses revealed that there was a significant difference between reported MET-minutes for leisure activities (*p* < 0.001) and walking (*p* < 0.001) between the three groups (oaBPPV_frail_> oaBPPV_robust_> controls_robust_). Both oaBPPV_frail_ and oaBPPV_robust_ reported significantly less moderate physical activity (*p* = 0.005) than control_robust_. The physical activity category also significantly differed (*p* < 0.001) between oaBPPV_frail_, oaBPPV_robust_ and controls_robust_ (Appendix A).

## 4. Discussion

This study aimed to compare objective and subjective measures of physical activity, frailty and well-being between oaBPPV and controls. OaBPPV were significantly less physically active. They performed significantly less active bouts and more sedentary bouts, and had a significantly different pattern of physical activity during the day compared to controls. OaBPPV reported significantly fewer minutes in physical activity for transports, leisure activities, walking and moderate-intensity activities. Their cognitive performance was significantly decreased and they were more (pre-)frail compared to controls. Moreover, oaBPPV_frail_ were even less physically active, more sedentary and experienced more fear of falling and feelings of depression compared to oaBPPV_robust_ and controls_robust_. Although more research is necessary to detangle the interaction between BPPV, physical activity, well-being and frailty, clinicians and researchers should consider that oaBPPV are less healthy and more at risk for frailty compared to controls. Therefore, it is recommended to conduct a follow-up assessments after treatment with repositioning maneuvers, not only to evaluate the resolution of BPPV but also to assess whether frailty has improved and physical activity has resumed, or if additional rehabilitation for frailty and physical activity is required.

The results indicate BPPV can induce a vicious cycle of fear of provoking symptoms, decreased physical activity, well-being and consequently frailty. In previous research, a decreased postural control [7,8], increased fear of falling and fall incidence [9] were also found in older adults with BPPV, reinforcing this vicious cycle leading to frailty.

However, it is also possible that frail and less physically active older adults have an increased prevalence of BPPV. It is hypothesized that, during sleep or prolonged sedentary behavior, otoconia accumulate or form an agglomerate. The decreased intensities seen in physical activity in the morning, and time-shift in physical activity during the day in oaBPPV can support this hypothesis. Possibly, regular movement can stimulate the dispersion of otoconia in the semicircular canals, and therefore reduce complaints. Future prospective research should investigate the prevalence of BPPV in older adults with a different frailty status and physical activity level.

It is noteworthy that both frailty and BPPV are more prevalent in women [11,35]. Also, there is growing evidence on the prevalence of decreased bone mineral density in patients with BPPV [36], suggesting a potential shared pathophysiological pathway with osteoporosis, which is highly prevalent in frail older adults [37].

Nevertheless, it is known that physical activity stimulates central adaptation mechanisms and enhances recovery in patients with vestibular disorders [14], and can serve as preventative strategy to slowdown/reverse frailty [38]. Although more research is necessary, clinicians should educate oaBPPV on the benefits physical activity for their recovery.

This need for education is also reflected in the discrepancy between the objectively and subjectively measured physical activity in oaBPPV. Although both were significantly decreased in comparison to controls, oaBPPV reported 64.9 MET-minutes/week of vigorous activity and the majority was classified as high physically active according to the IPAQ. However, objective measurements indicated that oaBPPV spend almost 70% of their dynamic time at a low and only 0.5% at a vigorous intensity. Although the IPAQ only inquires for physical activity with a minimal duration of 10 min, the MET-minutes reported were relatively high, whilst only 28% of the oaBPPV performed bouts of ≥10 min according to the MOX. This difference was smaller in controls, who reported 736.4 min of VPA/week but also spend 17.5% of their time at vigorous intensities according to the MOX. The discrepancy might be explained by recall bias in the IPAQ, as this induces a tendency to report experienced peaks in physical activity. Previous research has already demonstrated this recall bias in patients with vestibular disorders [16]. However, an objective assessment with accelerometer provides no information on perceived exertion. Possibly, oaBPPV experience light physical activities as moderate or vigorous, which should be assessed with a BORG Rating of Perceived Exertion Scale in future research.

The results of this study confirmed the limited existing literature on decreased physical activity and subjective well-being in patients with BPPV. To our knowledge, this is the first study taking the interaction with frailty into account. OaBPPV_frail_ were significantly less physically active and experienced more fear of falling and feelings of depression than oaBPPV_robust_ and controls_robust_. As both feelings of depression [39] and a reduced physical activity [14] are associated with residual dizziness after treatment with repositioning maneuvers, frailty might be an indicator for the need of more follow-up after treatment. Therefore, clinicians and researchers should be aware of the increased prevalence of (pre-) frailty in oaBPPV. Although it is not known whether frailty was already present before the presence of BPPV, screening and treating BPPV and educating patients on the importance of physical activity can prevent the aggravation from pre-frailty to frailty, and decrease the risk of adverse health outcomes. It is therefore recommended to perform a follow-up after treatment with the gold standard repositioning maneuvers. This follow-up should not only assess resolution of BPPV, but also assess whether frailty has improved and physical activities have been resumed, or if additional rehabilitation is necessary to recover frailty and physical activity.

This study has several limitations. By assessing physical activity with an accelerometer, the classification of intensities was based on an algorithm combining sensor orientation and accelerations, and bouts were interrupted when a 2-s change in classification was detected. The MOX was worn for four consecutive days and sleep was included in sedentary time, whereas the IPAQ inquires seven days and excludes sleep. Also, the IPAQ has been proven valid and reliable in healthy older adults, but not in (older) adults with vestibular disorders, which possibly limits score interpretation [40,41]. Frailty was not assessed with the gold standard measurement of Fried et al., [11] but was adjusted according to Àvila-Funes et al. [24] Also, frailty by Fried is focused on physical frailty, measures of psychosocial frailty were not included. The sub-analyses included small groups and no frail controls. The elaborative protocol and recruitment via outpatient care of the hospital may have caused (self-)selection bias among oaBPPV, as they needed to delay treatment and return to the hospital multiple times. Consequently, more mobile and less impaired patients may have been more likely to participate, possibly reducing the differences between patients and controls. Controls volunteering via public invitations may have also been biased, as they might be more physically and socially active. Nevertheless, this is the first study comparing both objectively and subjectively measured physical activity, frailty and subjective well-being between oaBPPV and controls matched for sex, age, weight and height, taking their differences in frailty into account. 

## 5. Conclusions

OaBPPV were significantly less physically active and more (pre-)frail than controls. They performed significantly less active bouts and more sedentary bouts, and have a different pattern of physical activity during the day. OaBPPV_frail_ were even less physically active, more sedentary and experienced more fear of falling and feelings of depression compared to oaBPPV_robust_ and controls_robust_. Future research should investigate whether physical activity, (pre-)frailty and well-being were already decreased before the BPPV onset, and if they recover after repositioning maneuvers, or if additional rehabilitation to recover these items is necessary. Nevertheless, clinicians treating oaBPPV are recommended to screen for frailty in oaBPPV and should educate the patients about the benefits of physical activity when treating oaBPPV. A follow-up should always be included after treatment with repositioning maneuvers in older adults. This follow-up should not only assess the resolution of BPPV, but also evaluate the presence of frailty and their physical activity, address their social and health implications and promote healthy aging in older adults.

## Figures and Tables

**Figure 1 jcm-13-07542-f001:**
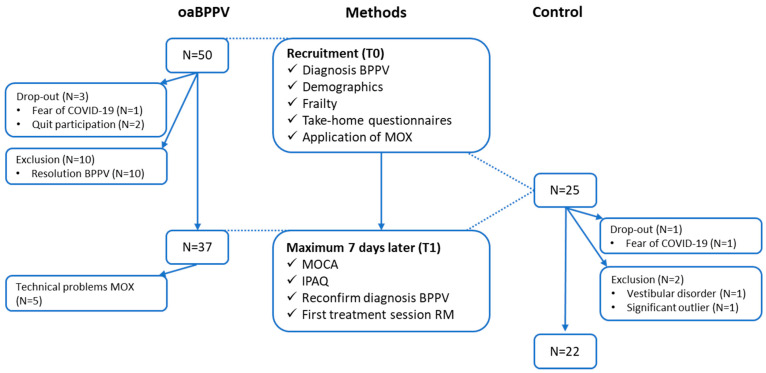
Flow charts of methods and selection process of participants. Abbreviations: oaBPPV, older adults with BPPV; Control, older adults in control group; BPPV, benign paroxysmal positioning vertigo; MOCA, montreal cognitive assessment; IPAQ, international physical activity questionnaire; RM; repositioning maneuver.

**Figure 2 jcm-13-07542-f002:**
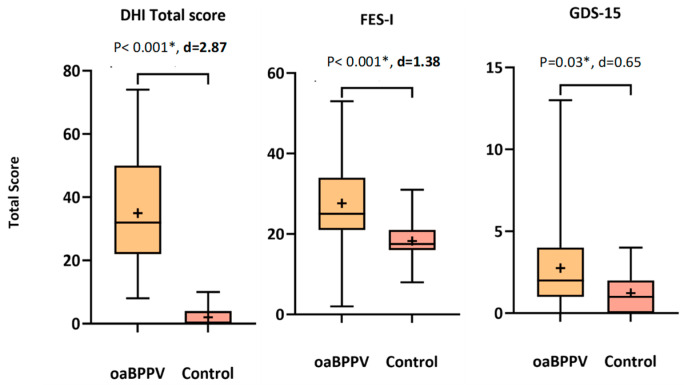
DHI, FES-I and GDS-15 scores in older adults with BPPV (*n* = 37) compared to an age-, weight-, and height-matched controls (*n* = 22). The boxplots indicate the medians, interquartile range and minimum and maximum values, with the’+’ indicating the mean values. Significant *p*-values are indicated with ‘*’. Sufficiently large Cohen’s d are indicated in bold. Abbreviations: oaBPPV, older adults with BPPV; Control, older adults in control group; DHI, dizziness handicap inventory; FES-I, falls efficacy scale international; GDS-15, 15-item geriatric depression scale.

**Figure 3 jcm-13-07542-f003:**
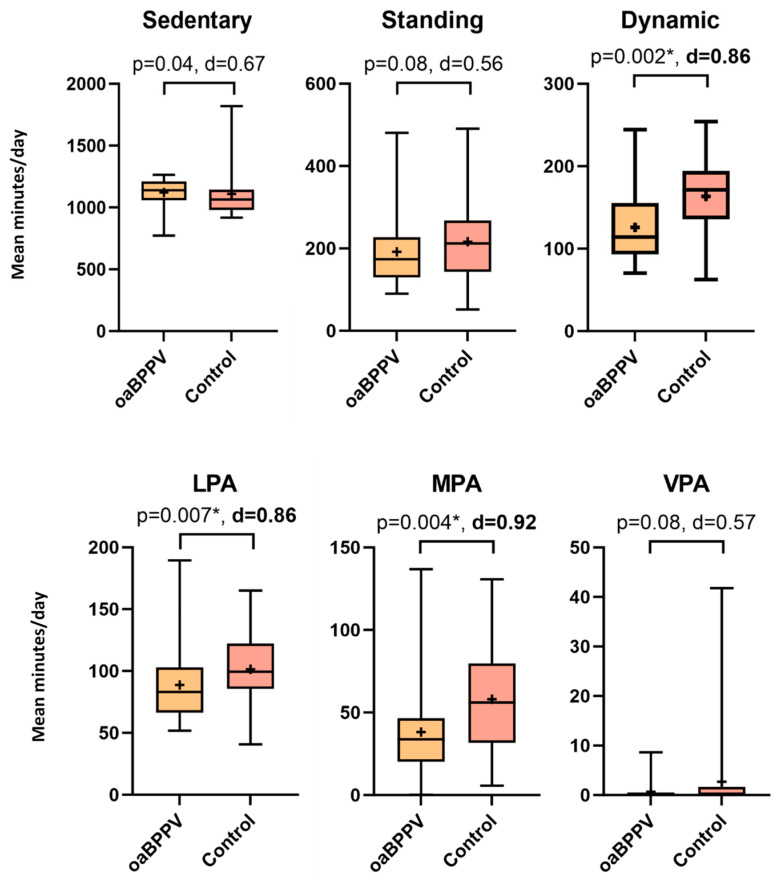
Mean minutes of physical activity classifications/day in older adults with BPPV (*n* = 32) compared to an age-, weight-, and height-matched controls (*n* = 22). The boxplots indicate the medians, interquartile range and minimum and maximum values, with the’+’ indicating the mean values. Significant *p*-values are indicated with ‘*’. Sufficiently large cohen’s d are indicated in bold. Abbreviations: oaBPPV, older adults with BPPV; Control, older adults in control group; LPA, low physical activity; MPA, moderate physical activity; VPA, vigorous physical activity.

**Figure 4 jcm-13-07542-f004:**
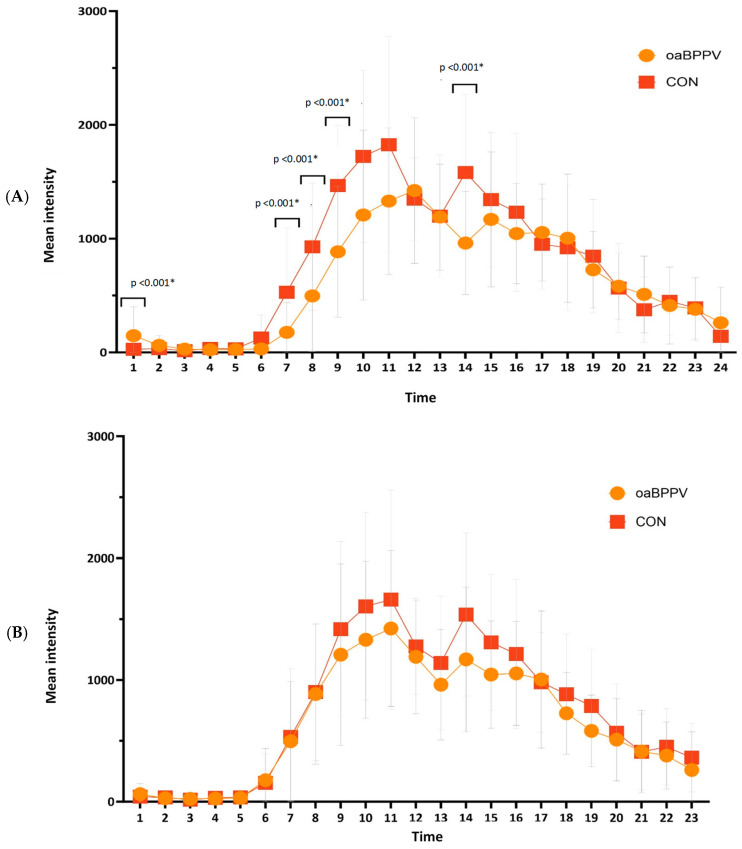
Distribution of the mean physical activity intensities per 24 h without (**A**) and with (**B**) 1-h time-shift in older adults with BPPV (*n* = 32) compared to an age-, weight-, and height-matched controls (*n* = 22). Significant *p*-values are indicated with ‘*’. Abbreviations: oaBPPV, older adults with BPPV; control, older adults in control group.

**Table 1 jcm-13-07542-t001:** Subject characteristics.

Characteristics	oaBPPV	Control	*p*-Value
N (F/M)	37 (23/14)	22 (12/10)	0.77
Age	73.1 (4.8)	73.5 (4.5)	0.77
Weight (kg)	76.8 (11.3)	73.6 (8.5)	0.25
Height (m)	1.7 (0.1)	1.7 (0.1)	0.16
BPPV			**<0.001**
RPSCC (*n*)	14	0	
LPSCC (*n*)	15	0	
Bilateral PSCC (*n*)	4	0	
RLSCC geotropic (*n*)/ apogeotropic (*n*)	1/3	0	
LLSCC geotropic (*n*)/ apogeotropic (*n*)	1/0	0	
No BPPV (*n*)	0	22	
Duration of complaints			**<0.001**
Some days (*n*)	3	0	
Several weeks (*n*)	5	0	
Several months (*n*)	29	0	
No complaints (*n*)	0	22	
Walking aid			0.19
None (*n*)	33	22	
Crutch (*n*)	3	0	
Walker (*n*)	1	0	
Sleeping pattern			0.21
Good (*n*)	20	14	
Restless (*n*)	12	5	
Long time needed to fall asleep (*n*)	2	2	
Restless + long time needed (*n*)	3	0	
Number of comorbidities	3 (0–6)	2 (0–5)	0.1
Number of medications	5 (0–11)	2.5 (0–10)	**0.007**
MOCA total score	23 (14–30)	27.5 (23–30)	**<0.001**

Significant differences are indicated in bold. Normally distributed data are expressed as mean (SD), non-normally distributed data as median (minimum-maximum). Abbreviations: oaBPPV, older adults with BPPV; control, older adults in control group; F, female; M, male; BPPV, benign paroxysmal positioning vertigo; RPSCC, right posterior semicircular canal BPPV; LPSCC, left posterior semicircular canal BPPV, RLSCC, right lateral semicircular canal BPPV; LLSCC, left lateral semicircular canal BPPV; MOCA, Montreal Cognitive Assessment scale.

**Table 2 jcm-13-07542-t002:** Frailty.

Frailty	oaBPPV	Control	*p*-Value	Cohen’s d
Robust (*n*) Prefrail (*n*) Frail (*n*)	10 15 11	17 4 0	**<0.001**	**1.27**
Unintentional weight loss Yes (*n*)/No (*n*)	7/29	3/18	0.46	0.37
Self-reported exhaustion Yes (*n*)/No (*n*)	18/18	0/21	**<0.001**	**1.22**
Slowness Yes (*n*)/No (*n*)	13/24	0/22	**0.001**	**0.92**
Weakness Yes (*n*)/No (*n*)	12/24	0/21	**0.002**	**0.86**
Physical inactivity Yes (*n*)/No (*n*)	7/29	1/20	**0.34**	**0.43**

Significant differences and sufficiently large cohen’s d are indicated in bold. Abbreviations: oaBPPV, older adults with BPPV; control, older adults in control group.

**Table 3 jcm-13-07542-t003:** Physical activity.

	oaBPPV	Control	*p*-Value	Cohen’s d
**Objective Physical Activity**				
**Number of bouts**				
Dynamic bouts 5–10 min	0.38 (0–4.5)	0.8 (0–3.5)	**0.02**	0.59
Dynamic bouts ≥10 min	0 (0–1.8)	0.5 (0–3)	**0.002**	**0.82**
Sedentary bout ≥30 min	10.2 (4.8–14.3)	6.7 (4.8–16)	**0.002**	**0.84**
**Bout duration (min)**				
Dynamic bouts ≥10 min	18.4 (10.5–27.5)	18.4 (10.5–27.5)	0.37	0.13
Sedentary bouts ≥30 min	56.8 (43.9–95.3)	55.9 (44.3–105.1)	0.4	0.09
**N with dynamic bouts ≥10 min** Yes/No	9/23	15/7	**0.002**	
**Subjective physical activity**				
**Categorical**				
Low (*n*) Moderate (*n*) Vigorous (*n*)	12 1 24	2 0 20	**0.04**	**0.61**
**MET-minutes/week**				
Work	0 (0–3360)	0 (0–10350)	0.05	0.26
Transport	0 (0–2799)	411 (0–2125.5)	**0.003**	**0.72**
Household	450 (0–4140)	705 (0–6342)	0.1	0.33
Leisure	360 (0–3226.5)	1548 (0–14697)	**<0.001**	**1.46**
Walking	132 (0–2079)	858 (0–7375.5)	**<0.001**	**1.09**
Sitting	369 (135.1)	350.5 (140.4)	0.24	0.22
MPA	720 (0–7020)	1770 (0–63720)	**0.004**	**0.72**
VPA	0 (0–960)	0 (0–9600)	0.07	0.32

Significant differences and sufficiently large Cohen’s d are indicated in bold. Normally distributed data are expressed as mean (SD), non-normally distributed data as median (minimum-maximum). Abbreviations: oaBPPV, older adults with BPPV; control, older adults in control group; MET, metabolic resting rate; MPA, moderate physical activity; VPA, vigorous physical activity.

## Data Availability

The data presented in this study are available on request from the corresponding author due to the nature of data (patient information).

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
