# Peer review of "Physical Activity and Frailty Are Impaired in Older Adults with Benign Paroxysmal Positional Vertigo"

_jcm, 2024, doi:10.3390/jcm13247542_

Round 1

Reviewer 1 Report

Comments and Suggestions for Authors

This is an interesting paper about a probable subgroup of BPPV patients. 

For the readers should be very useful to know why the patients go to the doctor i.e.: instability, dizziness, typical complaints of BPPV?

In other words, has this subgroup different symptom quality in comparison with "standard" BPPV patients?

Reviewer 2 Report

Comments and Suggestions for Authors

Thank you for the opportunity to review this well-constructed and insightful paper. This manuscript presents a relevant study addressing the relationship between physical activity, frailty, and BPPV in the elderly. Overall, it is a well-organized and methodologically sound paper; however, several clarifications are needed. Please consider the following points:

  1. Please write BPPV in full in the title.
  2. The study aims to compare objective and subjective measures of physical activity. The authors should clarify the motivation behind this choice, particularly in the context of BPPV. Expanding on the rationale will enhance understanding of the study’s aim.
  3. The methodology for confirming BPPV diagnosis through video Frenzel goggles requires further explanation. Please include a brief description of the principle behind this diagnostic tool, as well as how it operates in practice.
  4. The protocols for the Dizziness Handicap Inventory (DHI), Falls Efficacy Scale (FES-I), and the 15-item Geriatric Depression Scale (GDS-15) should be briefly expanded in the methods section. Additionally, it would be helpful to include information on their validity and reliability to strengthen the assessment of these tools.
  5. The manuscript currently defines sedentary activity as any time spent inactive (sitting/lying down). However, sedentary activity is distinct from inactivity. Please clarify this difference and ensure that your classification aligns with the existing literature on sedentary behavior.
  6. The authors state that dynamic minutes per day are categorized into light, moderate, and vigorous physical activity. The methodology for this classification requires clarification. Please provide details on the criteria or tools used to distinguish these intensities.
  7. A brief explanation of the items, components, scoring, validity, and reliability for the subjective well-being, MOCA, and IPAQ questionnaires should be provided. This will support the clarity and reproducibility of the methods.
  8. The IPAQ has been validated in a healthy population; however, using it to older persons with BPPV without assessing its validity and reliability in this setting may lead to complications in score interpretation. Kindly address this issue and incorporate this aspect as a limitation of the study.
  9. It may be beneficial to position the statistical analysis methods within a specific “Statistics” section to enhance the clarity of the methods.
  10.  Incorporating a “Participants” section at the outset of the Methods would improve the manuscript's organization. Furthermore, positioning the flowchart at the conclusion of this section would significantly enhance the clarity of the selection process and the study design.

We appreciate the efforts made by the authors in conducting this study and look forward to seeing these clarifications addressed.

Good luck

Reviewer 3 Report

Comments and Suggestions for Authors

Thank you for the opportunity to review your paper.

I found it very interesting.

I feel that your research has great social significance.

However, I was concerned that the important parts that serve as the basis for the study's conclusions were written in an abstract manner.

Details are provided below.

The definition of frailty in this study is vague.

The authors chose the Fried criteria, so I assume they want to focus on the relationship between BPPV and physical frailty.

Please clarify this point.

The items assessed by the authors include subjective well-being and depression.

These are because they have been noted to be related to mental/psychological frailty and social frailty.

Regarding the inclusion criteria, I believe that the two points "2) able to stand independently for at least 30 seconds" and "3) able to walk (with or without) walking aid for at least 10 meters" are more severe than those of older adults who exhibit physical frailty. Please provide the authors' reasons for this.

Regarding the exclusion criteria, it states "1) being unable to understand and follow simple instructions (e.g. due to severe dementia)," but older adults who have been diagnosed with dementia, even if they have mild dementia, are not called frail. Please explain your reasoning in detail.

The authors' conclusion, "It is necessary to determine whether further rehabilitation is needed," is non-specific. Please conclude with the implications of your research findings. In addition, please emphasize the novelty and social significance of your research.

That's all.

Round 2

Reviewer 3 Report

Comments and Suggestions for Authors

The authors have resubmitted the manuscript and appropriate revisions have been made.